# A Robust One-Step Recombineering System for Enterohemorrhagic *Escherichia coli*

**DOI:** 10.3390/microorganisms10091689

**Published:** 2022-08-23

**Authors:** Lang Peng, Rexford Mawunyo Dumevi, Marco Chitto, Nadja Haarmann, Petya Berger, Gerald Koudelka, Herbert Schmidt, Alexander Mellmann, Ulrich Dobrindt, Michael Berger

**Affiliations:** 1Institute of Hygiene, University of Münster, 48149 Münster, Germany; 2Institute of Food Science and Biotechnology, University of Hohenheim, 70599 Stuttgart, Germany; 3National Consulting Laboratory for Hemolytic Uremic Syndrome (HUS), 48149 Münster, Germany; 4Department of Biological Sciences, University at Buffalo, Buffalo, NY 14203, USA

**Keywords:** enterohemorrhagic *Escherichia coli*, recombineering, SOS response

## Abstract

Enterohemorrhagic *Escherichia coli* (EHEC) can cause severe diarrheic in humans. To improve therapy options, a better understanding of EHEC pathogenicity is essential. The genetic manipulation of EHEC with classical one-step methods, such as the transient overexpression of the phage lambda (λ) Red functions, is not very efficient. Here, we provide a robust and reliable method for increasing recombineering efficiency in EHEC based on the transient coexpression of *recX* together with *gam*, *beta*, and *exo*. We demonstrate that the genetic manipulation is 3–4 times more efficient in EHEC O157:H7 EDL933 Δ*stx1*/*2* with our method when compared to the overexpression of the λ Red functions alone. Both recombineering systems demonstrated similar efficiencies in *Escherichia coli* K-12 MG1655. Coexpression of *recX* did not enhance the Gam-mediated inhibition of sparfloxacin-mediated SOS response. Therefore, the additional inhibition of the RecFOR pathway rather than a stronger inhibition of the RecBCD pathway of SOS response induction might have resulted in the increased recombineering efficiency by indirectly blocking phage induction. Even though additional experiments are required to unravel the precise mechanistic details of the improved recombineering efficiency, we recommend the use of our method for the robust genetic manipulation of EHEC and other prophage-carrying *E. coli* isolates.

## 1. Introduction

Enterohemorrhagic *Escherichia coli* (EHEC) are zoonotic pathogens that can cause haemorrhagic colitis, haemolytic uremic syndrome (HUS), and eventually death [1,2]. EHEC have caused large foodborne outbreaks with thousands of infections worldwide that were associated with hundreds of hospitalizations and significant numbers of deaths [3,4,5]. The cardinal virulence factor of EHEC are Shiga toxins (Stx) that are encoded in lambdoid Stx prophages which inhibit eukaryotic protein synthesis by cleaving an adenine residue from the 28S ribosomal subunit [6].

All antibiotics were believed to increase *stx* expression [1], thereby resulting in an increased frequency of HUS [7]. Consequently, there is no causative cure for EHEC infections available. Nonetheless, it is clear that antibiotics that inhibit the bacterial gene expression apparatus do not enhance *stx* expression in vitro [8,9], and that these antibiotics were beneficial in a mouse infection model [10]. Regardless, the use of these antibiotics in therapy remains controversial.

Given these therapeutic limitations, it is necessary to develop alternative treatment options for EHEC infections. These developments will require a better understanding of the gene regulatory pathways that control pathogen behaviour. Therefore, efficient methods to genetically manipulate EHEC are essential. One of the most frequently applied site-directed mutagenesis methods in *Escherichia coli* (*E. coli*) is based on the overexpression of the phage lambda (λ) Red functions [11]. However, we and others had only sporadic success using the transient *gam*, *beta*, and *exo* overexpression system of Datsenko and Wanner [12]. In order to be able to more efficiently genetically modify EHEC, empirical improvements of the method, as well as more time-consuming two-step protocols have been developed [13,14].

We hypothesized that the process of transforming a linear PCR product, which is the essential step in the Datsenko and Wanner gene replacement protocol, may induce the bacterial SOS response. Induction of SOS would be especially problematic for genetically manipulating EHEC carrying Stx-encoding lambdoid Stx prophage, since the autoproteolysis of the CI-like repressors of lambdoid Stx prophages is stimulated by the coprotease activity of RecA, which is activated during the SOS response. Activated RecA [15] stimulated autocleavage of the CI-like repressor results in prophage and *stx* induction.

The formation of RecA nucleofilament on single-stranded DNA, and thereby activation of RecA depends on the activities of either two enzyme complexes, each of which respond to different types of DNA damage; RecBCD in the case of DNA double-strand breaks and RecFOR in the case of DNA gaps [16]. Neither RecA, nor RecBCD are necessary for recombineering using the λ Red functions [11,17]. Although the activation of RecA via RecBCD should be already sufficiently inhibited by Gam [18], the transformation of PCR products containing nicks or gaps could still activate RecA via the RecFOR pathway, resulting in the overall reduced efficiency of λ Red-based mutagenesis procedures.

The *E. coli* RecX protein is encoded in an operon with *recA*. It inhibits the coprotease activities of RecA in vitro and in vivo [19]. It also interacts with RecF [20] and its structure was used as the basis for the design of small molecule inhibitors of the SOS response [21]. We, therefore, decided to test if the transient coexpression of *recX* together with *gam*, *beta*, and *exo* would increase the frequency of recombinants in EHEC when compared to the *gam*, *beta*, and *exo* expression alone. We demonstrate here that the *recX* coexpression resulted in a significant 3–4-fold increase in the number of recombinants in the EHEC O157:H7 strain EDL933 Δ*stx1*/*2* (EHEC O157:H7 EDL933 Δ*stx1*/*2*), but not in the *E. coli* K-12 strain MG1655 (*E. coli* K-12 MG1655). The inhibition of a sparfloxacin-induced SOS response was not further enhanced in cells expressing *recX* in addition to *gam*, suggesting that the observed effect resulted from an inhibition of the RecFOR pathway.

## 2. Materials and Methods

### 2.1. Bacterial Strains and Plasmids Used in This Study

The bacterial strains used in this study are listed in Appendix A and the plasmids used in this study are listed in Appendix A.

### 2.2. Construction of Plasmids and Strains

The plasmid pKD46*recX* was constructed by amplifying the kanamycin resistance cassette from pKD4 [12] with primers MBP214 and MBP215 with Q5 polymerase (NEB, Ipswich, MA, USA) according to the manufacturer’s instructions. The resulting recombination substrate was integrated 3′ of *recX* in *E. coli* K-12 MG1655 with the help of pKD46 by Red recombineering [12]. *RecX*-kanR was afterwards amplified from the resulting strain by PCR primers MBP216 (which included the *gam*-*beta* spacer 5′-TAAAACGA-3′) and MBP217 with Q5 polymerase (NEB, Ipswich, MA, USA). The resulting PCR product was used to generate a recombination substrate with primers MBP218 and MBP219 with Q5 polymerase (NEB, Ipswich, MA, USA). This recombination substrate was afterwards used to transcriptionally fuse *recX* to *exo* of pKD46 by Red recombineering [12] in *E. coli* K-12 MG1655. The correctness of the *exo*-*recX* fusion was afterwards verified by Sanger sequencing (Eurofins Scientific SE, Luxembourg). The plasmid was thereafter named pKD46*recX*.

The plasmid pLP1 was constructed by removing the kanamycin resistance cassette of pKD46*recX* by digestion with XbaI (NEB, Ipswich, MA, USA) and ligation with T4 DNA ligase (NEB, Ipswich, MA, USA) after gel purification.

The plasmid pLP2 was constructed by amplifying *cat* of pMB54 [22] with primers MBPD80 (containing a PstI site) and MBPD81 (containing a KpnI site) and Q5 polymerase (NEB, Ipswich, MA, USA) according to the manufacturer’s instructions. The PCR product was afterwards digested with PstI (NEB, Ipswich, MA, USA) and KpnI (NEB, Ipswich, MA, USA) and, after purification, was cloned into a PstI (NEB, Ipswich, MA, USA) and KpnI (NEB, Ipswich, MA, USA) digested pUC18 [23] plasmid with T4 DNA ligase (NEB, Ipswich, MA, USA). The junction of the plasmid and *cat* was afterwards confirmed by Sanger sequencing (Eurofins Scientific SE, Luxembourg) and the plasmid was named pLP2.

The plasmid pWKSP*frr* was constructed by amplifying the promoter of *frr* (P*frr*) of *E. coli* K-12 MG1655 with primers MBP272 and MBP273 with Q5 polymerase (NEB, Ipswich, MA, USA) according to the manufacturer’s instructions. The PCR product was afterwards cloned into the SmaI (NEB, Ipswich, MA, USA) digested plasmid pWKS30 [24]. The plasmid was named pWKS*frr.*

The plasmid pWKSP*frr*-*cfp*-*aph*(3′)-*Ia* was constructed by amplifying *cfp* (cyan fluorescent protein)-*aph*(3′)-*Ia* with primers MBP274 and MBP275 and Q5 polymerase (NEB, Ipswich, MA, USA) and plasmid pMB47 [9] as a template. The fusion to P*frr* in pWKS*Pfrr* was then generated by Red recombineering [12] in *E. coli* K-12 MG1655. The junction of P*frr* and *cfp* was confirmed by Sanger sequencing (Eurofins Scientific SE, Luxembourg) and the plasmid was named pWKSP*frr*-*cfp*-*aph*(3′)-*Ia*.

*E. coli* K-12 MG1655 *att*::P*frr*-*cfp*-*aph*(3′)-*Ia* was constructed by amplifying P*frr*-*cfp*-*aph*(3′)-*Ia* from the plasmid pWKSP*frr*-*cfp*-*aph*(3′)-*Ia* with primers MBP276 and MBP277 and Q5 polymerase (NEB, Ipswich, MA, USA) according to the manufacturer’s instructions. The integration of the recombination substrate in the *attB* site of *E. coli* K-12 MG1655 was achieved with Red recombineering [12]. The junctions to the chromosome were afterwards verified by PCR with primers MBP280 and MBP273 (5′ junction) and MBP278 and MBP279 (3′ junction). The PCR products were in addition verified by Sanger sequencing (Eurofins Scientific SE, Luxembourg).

*E. coli* K-12 MG1655 P*recA* SENSOR was constructed by amplifying P*recA*-*yfp* (yellow fluorescent protein)-*cat* with primers MC204 and MC205 and Q5 polymerase (NEB, Ipswich, MA, USA) and pMBM22 [25] as a template. This recombination substrate was integrated convergent to the P*frr*-*cfp* cassette in *E. coli* K-12 MG1655 *att*::P*frr*-*cfp*-*aph*(3′)-*Ia* by Red recombineering [12]. The junction of the chromosome with the P*recA*-*yfp* cassette was confirmed by PCR and the PCR product was verified by Sanger sequencing (Eurofins Scientific SE, Luxembourg). The strain was thereafter named *E. coli* K-12 MG1655 P*recA* SENSOR.

### 2.3. Comparison of the Recombination Efficiency of pKD46recX with pKD46recX

A detailed protocol for the generation of the recombination substrates and a step-by-step protocol for the preparation of recombination proficient competent cells are provided in the Appendix A. Briefly, pKD46 and pKD46*recX* were transformed in *E. coli* K-12 MG1655 and EHEC O157:H7 EDL933 Δ*stx1*/*2* and glycerol stocks were made. For each experiment, the strains were plated on LB plates containing ampicillin [100 µg/mL] and incubated at 30 °C overnight. The next day, a single colony was picked and inoculated in 2 mL dYT containing ampicillin [100 µg/mL] and incubated at 180 rpm at 30 °C overnight. The next day, the overnight cultures were diluted 1:200 in 100 mL dYT containing ampicillin [100 µg/mL] and incubated at 180 rpm at 30 °C. When the OD_600nm_ reached 0.4, 10% L-arabinose solution was added to a final concentration (f. c.) of 0.3% and the incubation was continued at 180 rpm at 37 °C for 1 h. Afterwards, the bacteria were chilled on ice, harvested by centrifugation, and washed three times with ice-cold water. The pellet was resuspended in a small volume of ice-cold water and the OD_600nm_ of a 1:100 dilution was determined. Next, the bacterial suspension was adjusted to an OD_600nm_ = 45 − 54. Seventy µL of this suspension was used to transform four aliquots of the indicated amounts of the recombination substrates (stored at −20 °C) by electroporation (Bio-Rad Micropulser, settings 1.8 kV/6 ms). Afterwards, the bacteria were washed out of the cuvette with 1 mL dYT medium and incubated for 1 h at 37 °C with shaking. Afterwards, the bacteria were harvested by centrifugation, resuspended in 100 µL fresh dYT and spread on an LB plate containing chloramphenicol [12.5 µg/mL]. The plates were incubated overnight at 37 °C and the next day the colonies per plate were counted. The technical replicate that deviated the most from the average of colony numbers was excluded and the remaining three technical replicates were used for further evaluation. From these plates, ten randomly chosen colonies were used for the PCR analysis (for the details see the Appendix A). The whole procedure was repeated three times for each recombination substrate.

### 2.4. Test of pKD46recX in E. coli O104:H4

The plasmid pKD46*recX* was transformed into electrocompetent *E. coli* O104:H4 cells and recombination proficient cells were prepared (see Appendix A, step-by-step protocol). In order to replace *stx2*, a recombination substrate was generated with primers MBP223 and MBP364 and pLP1 as a template using Q5 polymerase (NEB, Ipswich, MA, USA) and purified as described for the other recombination substrates described above. The recombination substrate was transformed in the recombination proficient *E. coli* O104:H4 cells and after one-hour recovery at 37 °C, the cells were spread on LB agar plates containing chloramphenicol [12.5 µg/mL] and incubated overnight at 37 °C. The replacement of *stx2* by *cat* was afterwards confirmed by PCR with primers MBP226 and MBP260 (5′ junction) and MBP5 and MBP227 (3′ junction) and whole-genome sequencing (PacBio, Menlo Park, CA, USA).

### 2.5. Measurement of SOS Response

Plasmids pKD46 and pKD46*recX* were transformed in the SOS response reporter strain *E. coli* K-12 MG1655 P*recA* SENSOR. The bacteria were grown overnight in the M9 medium supplemented with 0.4% glucose containing ampicillin [100 µg/mL] at 30 °C and 180 rpm in an INFORS HT Multitron incubator. The next day, the bacteria were diluted 1 to 10 in M9 medium supplemented with 0.4% arabinose or 0.4% glucose, 0.4% casamino acids and [100 µg/mL ampicillin] (150 µL final volume) in black µ-clear plates with a transparent bottom and lid (Greiner Bio-One, Frickenhausen, Germany) and incubated in a Tecan infinite 2000pro instrument at 30 °C with shaking. Sparfloxacin [15 µg/mL f. c.] was added after 1 h of incubation. Every ten minutes the optical density was automatically recorded at 595 nm and the YFP signal was recorded with excitation at 514 nm, emission at 550 nm, and with a gain 100. The optical density was corrected for a medium blank and the fluorescence signal for background fluorescence (*E. coli* K-12 MG1655 transformed with pKD46). The corrected fluorescence signal was normalized to the corrected optical density and the promoter activity of P*recA* was calculated as an increase of the normalized fluorescence signal (F/OD) over time: d(F/OD) = (F/OD)_n_ − (F/OD)_n−1_. Negative values for d(F/OD) were set to zero. The overall SOS response was calculated by F/OD response = F/OD (250 min) − F/OD (60 min) for each biological replicate. The average values for the overall SOS response were calculated by using three biological replicates.

### 2.6. Statistics and Data Visualization

Statistical analysis was performed using GraphPad Prism 8.0.1 statistical analysis software (www.graphpad.com, accessed on 12 May 2022). A Student’s t-test was used with a *p*-value < 0.05 considered as statistically significant. Figures were created in R v.4.0.3 computing environment (R Core Team, 2021) using R package ggplot2 (Wickham, 2016) and Inkscape v1.1 (Inkscape Project, 2020). DNA sequence alignments were completed with CLC Workbench 6.

## 3. Results

### 3.1. Construction of Plasmids

The plasmid pKD46*recX* was constructed by transcriptionally fusing *recX* to the *exo* gene in pKD46 [12] by λ Red recombineering. In order to ensure the coexpression of *recX* with *gam*, *beta*, and *exo*, the gam-beta spacer (5′-TAAAACGA-3′) was introduced 5′ of the ATG of *recX* (for the details see Materials and Methods). The plasmid pLP2 was constructed by cloning *cat* of pMB54 [22] into the multiple cloning site of pUC18 [23]. The plasmid pLP2 was used for the generation of all chloramphenicol resistance cassette (*cat*) encoding recombination substrates that were used in this work.

### 3.2. Selection of Conserved Chromosomal Loci for a Direct Comparison of Recombination Efficiency in EHEC O157:H7 EDL933 Δstx1/2 and E. coli K-12 MG1655 and Generation of Recombination Substrates

We wanted to directly compare the effect of pKD46*recX* on recombination efficiencies in an EHEC background vs. *E. coli* K-12 MG1655. For safety reasons, we used an *stx* negative derivative of EHEC O157:H7 EDL933 for these experiments [26]. In order to make the relevant comparisons, we needed to use the same recombination substrates and targets in both genetic backgrounds, meaning we needed to identify genetic loci that were conserved on the nucleotide level in between both strains. We identified chromosomal regions of perfect chromosomal homology (CH) to design primers with 50 bp overhangs in order to delete *lacI* and *lacZYA* in EHEC O157:H7 EDL933 Δ*stx1*/*2* as well as in *E. coli* K-12 MG1655 (Appendix A). We also wanted to test the effect of pKD46*recX* on the efficiency of inserting the *cat* gene into the chromosome. The region upstream of *attB* (with respect to the direction of replication) is highly conserved on the nucleotide level between both strains (Appendix A). Therefore, we targeted the chromosomal region upstream of *attB* for the design of a recombination substrate. The positions of the CH with respect to *lacI*, *lacZYA* and *attB* are graphically depicted in Figure 1.

The recombination substrates for the deletion of *lacI* and *lacZYA* and the insertion upstream of *attB* were generated by PCR using the primers listed in Appendix A and pLP2 as templates. This first PCR product was then purified from an agarose gel and served as a template for a second PCR to generate the recombination substrates with the same primers. The recombination substrates were analysed qualitatively by agarose gel electrophoresis, quantified, and afterwards stored in aliquots at −20 °C until use in the experiments.

### 3.3. Test for the Optimal Induction Time of the λ Red Functions

Expression of the λ Red genes, *gam*, *beta*, and *exo*, which are required for recombineering, is induced by adding arabinose during the preparation of electrocompetent target cells bearing the pKD46 plasmid [12]. The prolonged expression of the λ Red functions is mutagenic [13]. However, expressing these functions for too short a time may have reduced the recombination frequency. Therefore, to ensure we identified the best possible conditions for recombination, we determined the optimal time of induction. For these experiments, pKD46 [12] containing *E. coli* K-12 MG1655 cells were exposed to 0.3% arabinose for various time periods and then electrotransformed with 0.5 ng of the *lacZYA* locus recombination substrate (Figure 1A). We obtained the highest number of colonies after 1 h of induction of the expressions of *gam*, *beta*, and *exo* (Appendix A). Longer induction times were deemed not to be needed since 1 h of exposure to the arabinose inducer provided both adequate numbers of transformants while not increasing off-target mutations [13]. All of the subsequent experiments were therefore completed with 1 h of induction time.

### 3.4. Comparison of the Numbers of Recombinants in E. coli K-12 MG1655 and EHEC O157:H7 EDL933 Δstx1/2

Preliminary observations indicated that, for any given locus, genetic modification of EHEC O157:H7 EDL933 Δ*stx1*/*2* chromosome required substantially higher amounts of the same recombination substrate than *E. coli* K-12 MG1655. The amounts of recombination substrate needed to successfully modify the chromosome also varied between the chromosomal loci within each individual genetic background. Therefore, in order to obtain appropriate numbers of recombinants on selective agar plates for enumeration, the amounts of recombination substrate were adjusted to genetic background and chromosomal position. For the details, see the Materials and Methods section (a detailed step-by-step protocol is additionally provided in the Appendix A). Figure 2A shows that at each chromosomal locus, we obtained significantly higher numbers of colonies (3–4-fold) in EHEC O157:H7 EDL933 Δ*stx1*/*2* bearing pKD46*recX* compared to cells bearing pKD46 (the technical replicates for each biological replicate are shown in Appendix A). In contrast, the effect of expressing *recX* from pKD46 on recombination efficiency varied between the three individual targeted *E. coli* K-12 MG1655 loci. In this strain, only the integration of the resistance cassette upstream of *attB* was significantly more efficient when pKD46*recX* was used, albeit less than two-fold (Figure 2B; Appendix A shows the technical replicates for each biological replicate).

To determine the accuracy of recombination, we performed a PCR analysis to determine the location of the insertion or deletion in both EHEC O157:H7 EDL933 Δ*stx1*/*2* and *E. coli* K-12 MG1655. For these experiments, we picked 10 random colonies for each biological replicate, plasmid, and chromosomal locus and performed a PCR analysis (Appendix A). The PCR analysis indicated that the resistance cassette was integrated at the correct chromosomal locus in both genetic backgrounds in almost all cases.

In addition to that, we have also used pKD46*recX* to replace *stx2* of *E. coli* O104:H4 by *cat* (Appendix A; see Materials and Methods).

### 3.5. Coexpression of RecX with Gam, Beta, and Exo Does Not Further Repress the Sparfloxacin-Dependent SOS Response Induction

Sparfloxacin is a quinolone antibiotic that inhibits DNA gyrase [27,28], thereby causing double-stranded breaks and ultimately arresting DNA replication and activating RecA [29]. To investigate the mechanism by which *recX* impacts recombination efficiency, we decided to test if the coexpression of *recX* together with the λ Red functions would further repress a sparfloxacin-induced SOS response. For these experiments, we constructed an *E. coli* K-12 MG1655 strain with a *recA* promoter (P*recA*)-*yfp* transcriptional fusion chromosomally integrated into *attB* (*E. coli* K-12 MG1655 P*recA* SENSOR; for the details, see Materials and Methods). Subsequently, the reporter strain was transformed separately with plasmids pKD46 and pKD46*recX*. The encoded enzymes in both pKD46 and its derivative pKD46*recX* are under the control of the *araBAD* promoter. This promoter is activated by the addition of arabinose to the growth medium as described above but strongly repressed by the addition of glucose [30]. We measured the effect of sparfloxacin addition on the P*recA* fluorescent reporter activity in cells grown in M9 + glucose (uninduced) or M9 + arabinose (induced).

When *E. coli* K-12 MG1655 P*recA* SENSOR transformed with either pKD46 or pKD46*recX* were grown with glucose, the fluorescence signal started to increase shortly after sparfloxacin was added to the medium (black arrow) and the signal further increased until ~250 min. This demonstrated that P*recA* was activated by sparfloxacin, as expected (Figure 3, filled symbols) [27]. In contrast, when these cells were grown in the presence of arabinose, the increase in the fluorescence signal in response to the sparfloxacin addition was substantially lower (Figure 3, open symbols) in both pKD46 and pKD46*recX* strains. The coexpression of *recX* together with *gam*, *beta*, and *exo* (Figure 3, open diamonds) did not further reduce the sparfloxacin-induced SOS response when compared to the expression of *gam*, *beta*, and *exo* alone (Figure 3, open circles). In contrast, the coexpression of *recX* slightly, but significantly attenuated the SOS response repression by Gam, as judged by the overall response of P*recA* (Appendix A).

## 4. Discussion

EHEC infections still constitute a significant worldwide health problem [3,4,5,31]. Even though the use of appropriate antibiotics in therapy is recently again more frequently discussed [2,9,32], it remains controversial [7,33]. In order to develop alternative therapy options, a better understanding of EHEC is essential. However, the genetic manipulation of EHEC with classical methods has only achieved sporadic success. To overcome this problem, several other groups have relied on empirically optimized or more time-consuming two-step recombineering methods [13,14]. The goal of our work was to provide a reliable method for increasing recombineering efficiency in EHEC strains. Even though also other important factors, e.g., the overexpression system itself are known to affect recombination efficiencies and may also further improve our system (for review, see [34]), we used pKD46 as the starting point for our investigations, as it is one of the most commonly used plasmids for recombineering [12,34].

Figure 2 shows that genetic modification of three conserved chromosomal loci in EHEC O157:H7 EDL933 Δ*stx1*/*2* was 3–4 times more efficient in cells coexpressing *recX* together with *gam*, *beta*, and *exo*, as compared to the same strain that only expressed the λ Red functions. Importantly, we found that recombination took place accurately, regardless of locus and whether the product of the reaction was an insertion or deletion, demonstrating that this coexpression did not increase the frequency of off-target mutagenic recombination. Significantly, in addition to EHEC O157:H7 EDL933 Δ*stx1*/*2*, we have also used pKD46*recX* to successfully modify clinically relevant EHEC wild-type strains, strains in which successful recombineering was rather sporadic. For example, we used pKD46*recX* to delete the *stx2* gene of the highly virulent *E. coli* O104:H4 [5].

The mechanism by which coexpression of *recX* with *gam*, *beta*, and *exo* increased recombination efficiency in EHEC O157:H7 EDL933 Δ*stx1*/*2* is, so far, not clear. We considered several possibilities. First, it is possible that the higher number of chloramphenicol-resistant colonies obtained when using pKD46*recX* in EHEC O157:H7 EDL933 Δ*stx1*/*2* may not result from an increased recombination efficiency, but instead result from an improvement in transformation efficiency. However, we routinely tested each batch of competent cells by transforming 0.5 ng of pLP2 and plating an aliquot of the transformation reaction. We did not observe an improvement in the transformation efficiency. Instead, we observed the opposite, i.e., when pKD46*recX* was used, the transformation efficiency dropped in both *E. coli* K-12 MG1655 and to a lesser degree in EHEC O157:H7 EDL933 Δ*stx1*/*2*. This drop in transformation efficiency following *recX* overexpression was expected since activation of RecA is known to increase the transformation efficiency [35]. However, the number of recombinants in EHEC O157:H7 EDL933 Δ*stx1*/*2* increased, while the recombination efficiency in *E. coli* K-12 MG1655 did not drop to the same degree as the transformation efficiency in this strain. Therefore, either the transformation efficiency of linear DNA was less affected by *recX* expression, or the drop in transformation efficiency was outweighed by a concomitant general increase in recombination efficiency.

We speculated that electroporation of linear DNA into cells may activate RecA and the bacterial SOS response impacting, directly or indirectly, recombination frequency, thereby activating lytic growth of prophage resident within EHEC strains, eliminating them from the population of possible recombination targets. This suggestion is consistent with the observation that expression of *recX* together with λ Red function had no significant impact on recombination efficiency in *E. coli* K-12 MG1655, a strain that lacks inducible prophage (Figure 2; Appendix A).

Depending on the type of DNA damage, the SOS response and RecA activation can be stimulated by either or both of two enzyme complexes RecBCD and RecFOR. Quinolone antibiotics stimulate the SOS response via RecBCD and Gam protein inhibits the activation of RecA by interacting with the DNA binding site of the RecBCD complex [18]. If RecX affected SOS activation and thereby recombineering efficiency via the RecBCD pathway, coexpression of this protein would be predicted to further reduce Gam inhibition of *recA* promoter activity. However, *recX* coexpression did not cause further inhibition of the quinolone-induced SOS response (Figure 3; Appendix A), indicating that this potential pathway of SOS response induction by the transformation of a linear recombination substrate is already sufficiently blocked by *gam* expression alone. This finding suggests that RecX does not act to increase recombination via inhibition of the RecBCD pathway.

The RecFOR pathway of the SOS response induction, however, is not blocked by Gam. This pathway has a role in mediating the recombination of damaged plasmid DNA with the bacterial chromosome [36]. Given our findings, we suggest that the process of transformation of recombination substrates may induce the RecFOR pathway of SOS induction, especially when mechanically damaged DNA or DNA damaged by the freezing–thawing process is used [37]. In this case, the increase in recombination efficiency in EHEC O157:H7 EDL933 Δ*stx1*/*2* that was observed here might be indeed due to the RecX-dependent inhibition of the RecFOR pathway of SOS response induction and the prevention of subsequent phage-mediated lysis [15,20]. Despite the attractiveness of this idea, further experiments are necessary to clarify the exact mechanism. Nevertheless, we have already used pKD46*recX* successfully in clinical non-EHEC *E. coli* isolates in which we have failed to construct mutants with pKD46. Whether or not this is due to the presence of uncharacterised SOS-inducible prophages as well, is not clear. However, it indicates that using pKD46*recX* might be of advantage in case the phage content of a strain is not known [38].

We also found that the amount of recombination substrate needed to obtain enumerable numbers of recombinant colonies varied between chromosomal loci and genetic backgrounds. Interestingly, we observed that insertion at the *attB* locus required the addition of the highest amount of recombination substrate, regardless of genetic background and even though the product of the reaction resulted in a small, ten base pair deletion. The *attB* locus is more distant from the origin of replication than *lacI* and *lacZYA*. This finding is consistent with earlier work demonstrating that phage P1 transduction efficiency also decreases with increasing distance of the target from the origin of replication [39]. Since logarithmically growing cells are used for recombineering, this may therefore simply reflect the general relative increase of the copy numbers of the target loci, or a relatively higher number of replication forks at the specific locus, which are required for Red recombination [40]. However, a more systematic investigation would be necessary to clarify the question if a phage P1-like bias also exists for recombineering.

## 5. Conclusions

The work presented here describes a robust and easy-to-implement strategy for increasing the efficiency of genetically engineering clinically important *E. coli*. Even though some elements of the pKD46*recX* recombineering system and of its derivative pLP1 (lacking the kanamycin resistance) may be subject to further optimization [34], even the relatively modest, but robust increases in recombination efficiency in EHEC or any other hard-to-manipulate *E. coli* background facilitated by this strategy clearly can make a difference between failure and success.

## Figures and Tables

**Figure 1 microorganisms-10-01689-f001:**
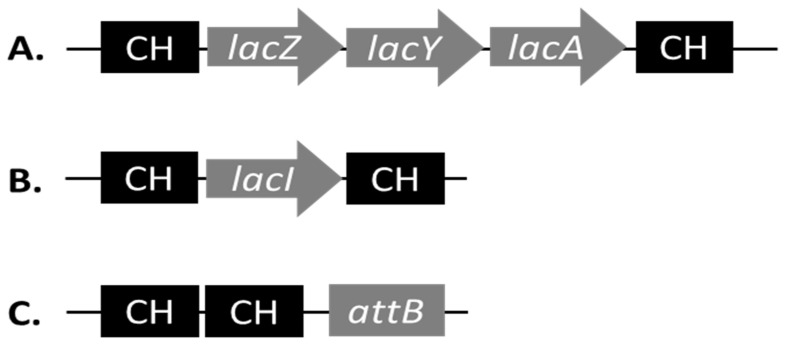
Schematic representation of the positions of chromosomal homology (CH) between EHEC O157:H7 EDL933 and *E. coli* K-12 MG1655 that were used to design the primers for the recombination substrates. (**A**) CH for the deletion of the *lacZYA* operon. (**B**) CH for the deletion of the regulatory gene *lacI*. (**C**) CH for the insertion upstream of the phage lambda attachment site *attB*.

**Figure 2 microorganisms-10-01689-f002:**
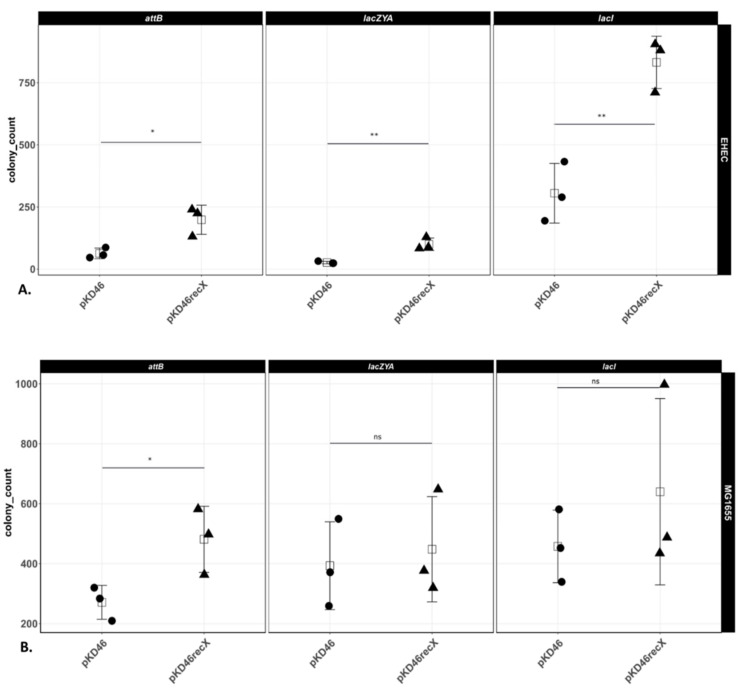
Numbers of chloramphenicol-resistant colonies obtained for the indicated genetic locus. (**A**) Number of chloramphenicol-resistant colonies in EHEC O157:H7 EDL933 Δ*stx1*/*2*. (**B**) Number of chloramphenicol-resistant colonies in *E. coli* K-12 MG1655. Shown are the average values of three technical replicates of three independent experiments. Statistically significant t-test comparisons are indicated by * (*p* < 0.05), ** (*p* < 0.01) and “ns” if not statistically significant.

**Figure 3 microorganisms-10-01689-f003:**
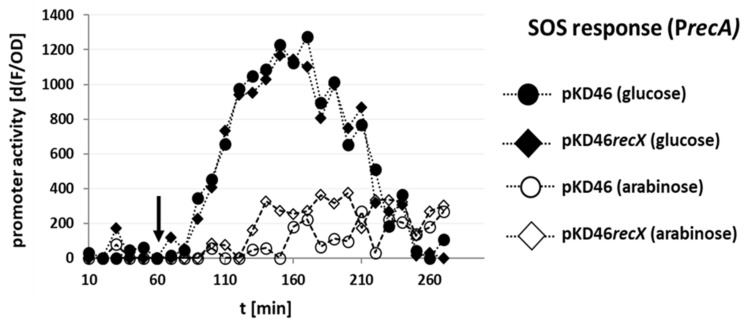
Sparfloxacin-dependent induction of the SOS response in *E. coli* MG1655 P*recA* SENSOR. The promoter activity of P*recA* as the increase of fluorescence over time in the presence of the indicated plasmid and the indicated sugar is shown. The black arrow indicates the time point of the addition of sparfloxacin (f. c. 15 µg/mL).

## Data Availability

The data presented in this study are available in article and Appendix A.

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
