# Peer review of "A Robust One-Step Recombineering System for Enterohemorrhagic *Escherichia coli"

_microorganisms, 2022, doi:10.3390/microorganisms10091689_

Round 1

Author Response

We thank the reviewer for the valuable comments. Answers in italic.

 The manuscript entitled: A robust one-step recombineering system for enterohemorrhagic Escherichia coli by Lang Peng and co-authors concerns the important problem of searching for alternative EHEC treatment options. The novelty consists in the use of the pKD46recX construct to modify EHEC strains. EHEC genetic manipulation is a promising method on the way to developing alternative therapeutic options for both these strains and other transmitting prophages. The results are well documented and discussed.

A few minor remarks that apply to the editorial side of the work:

  1. Throughout the text, the citations should be corrected following the instructions of the authors

We have done this.

 Please pay attention to typos and repetitions:

- line 49 – “geneticalsly” 

We have corrected this (and other typos).

- line 121 - "by Red recombineering"

We have corrected this.

- line 93 - correct the record of the spacer sequence

We thank the reviewer for noting that. We have corrected the sequence.

 - line 154 - correctly write the OD value

We have corrected this.

Reviewer 2 Report

The main finding of this study was the development of a more efficient recombination system for Enterohemorrhagic Escherichia coli (EHEC). The current methods to generate gene knockouts in EHEC strains are difficult and less than efficient due to the number of lysogenic prophages. The SOS response to a deleterious injury to the cell can cause these lysogenic prophages to become lytic resulting in the lysis of the cell. This new method seeks to repress the prophage induction. This new method should result in additional studies that use genetic manipulation to identify protein functions in this important group of human pathogens.

The strength of this manuscript is the amount of work that was accomplished making the plasmid constructs, comparing recombination efficiency at different locations, and verifying the recombination events in the strains. The manuscript could be strengthened if there was a figure that showed the genetic manipulation used to make the plasmids. I found it hard to follow some of the plasmid construction without a diagram to follow.

The molecular methods used to make the plasmids in the manuscript are well established in the literature. The results are most presented in a straightforward easy to understand progression. The results are supported by the data.

Detailed review.

Line 35. I think “food-born” should be “foodborne”

Line 49. Genetically is spelled wrong.

Line 62. Maybe think about changing the sentence to read “…results in prophage and Stx induction.” I know that prophage induction is the key component. However, this would also show the importance to causing disease.

Line 65. The sentence should read ..of either..”.

Lines 98 – 137. There are four sentences where “controlled” is used to describe the sequencing results. I would suggest using the word “verified”

Line 120. I didn’t see plasmid pMB47 in Supplementary Table 2.

Line 121. “Red recombineering” is used twice back-to-back. It only needs to be used once.

Line 155. I am not sure what an “OD660nm =45 -54” is supposed to signify.

Line 183. Please define what is meant by an “.. infors incubator.”

Line 207. I am not sure “transcriptionally” is the right work. I would suggest something like “.. by fusing recX in frame with the exo gene ..”

There are a lot of gene names that are not italicized in the manuscript.  Also, E. coli is not italicized after the Results section.

226. Please change “..genes..” to read “.. of inserting the cat gene into the chromosome.”

Figure 1 legend. Remove “..in..” from “ ..in between..”

Line 256. I think the citation is not in superscript for “..mutations13.”  There are other places in the manuscript where this same error is occurring.

Faithfulness

Line 282. Change the sentence to read “..we have also used pKD46recX to replace the stx2 gene of E .coli O104:H4 with cat.

Line 282. This result seems to be an afterthought. There is only one sentence for the result with no verification that stx2 gene was inactivated. This result is barely mentioned in the Discussion (line 344). I like that lab adapted strains were used to develop the method, but the real result here is the application of the method to a wild type EHEC strain. This should be one of the main points of the manuscript.

Line 293.  “ultimately” should only be used once in the sentence.

Line 307. Please indicate what fluorescent signal started to increase.

Line 317. The sentence reads awkwardly “..co-expression of recX apparently slightly, but significantly attenuated..” Either the result is significant or not.

Supplemental Figure 6. Please add E. coli K-12 MG1655 to the Figure or the Figure legend.

Lines 340-342. I missed how you determined there were no off-target mutagenic recombination. Did you sequence the recombinant strains?

Line 345. Change the sentence to read “..to delete the stx2 gene of the highly..”

Line 348.  Change “in” to “of”.

Line 365.  I think you can remove “prophage containing” from the sentence.

Line 392. Remove “also” from the sentence.

Supplementary Figures 3 and 4.  Please add to the Figure legend what type of plot this is with the line representing the standard deviation and the box the average of the three replicates. Also, describe what the asterisks and “ns” indicate.

Supplementary Figure 5 legend. Remove “in” before “between”

Author Response

We thank the reviewer for the valuable comments. Answers in italic.

The main finding of this study was the development of a more efficient recombination system for Enterohemorrhagic Escherichia coli (EHEC). The current methods to generate gene knockouts in EHEC strains are difficult and less than efficient due to the number of lysogenic prophages. The SOS response to a deleterious injury to the cell can cause these lysogenic prophages to become lytic resulting in the lysis of the cell. This new method seeks to repress the prophage induction. This new method should result in additional studies that use genetic manipulation to identify protein functions in this important group of human pathogens.

The strength of this manuscript is the amount of work that was accomplished making the plasmid constructs, comparing recombination efficiency at different locations, and verifying the recombination events in the strains. The manuscript could be strengthened if there was a figure that showed the genetic manipulation used to make the plasmids. I found it hard to follow some of the plasmid construction without a diagram to follow.

We are sorry for that. We did not provide a construction scheme, as we did not want to distract from the main message of the paper. However, the relevant parts of the plasmid are sequenced and the plasmid is available upon request.

The molecular methods used to make the plasmids in the manuscript are well established in the literature. The results are most presented in a straightforward easy to understand progression. The results are supported by the data.

Detailed review.

Line 35. I think “food-born” should be “foodborne”

We have corrected this.

Line 49. Genetically is spelled wrong.

We have corrected this.

Line 62. Maybe think about changing the sentence to read “…results in prophage and Stx induction.” I know that prophage induction is the key component. However, this would also show the importance to causing disease.

We agree with the reviewer that this sentence makes it easier for the reader to understand that the same mechanism that is essential for disease development may also cause the problem in the genetic manipulation of EHEC. We have changed the sentence.

Line 65. The sentence should read ..of either..”.

We have corrected this.

Lines 98 – 137. There are four sentences where “controlled” is used to describe the sequencing results. I would suggest using the word “verified”

We have corrected this.

Line 120. I didn’t see plasmid pMB47 in Supplementary Table 2.

We thank the reviewer for noting this. We have introduced pMB47 in Table 2.

Line 121. “Red recombineering” is used twice back-to-back. It only needs to be used once.

We have corrected this.

Line 155. I am not sure what an “OD660nm =45 -54” is supposed to signify.

It is the calculated OD600nm value of a concentrate of a typical bacterial suspension. If 100 ml of a bacterial suspension of an OD600nm = 0.4 is concentrated in 0.5 - 1 ml, an OD600nm = 45 -54 is a rather typical value that one obtains (that is of course not measured directly, but in a 1:100 dilution).

In order to be able to make comparisons in between batches and bacterial strains we recommend adjusting the OD at the end of the preparation of e.c.c. at any instance (especially, if clinical isolates with different sedimentation properties during centrifugation are used).

Line 183. Please define what is meant by an “.. infors incubator.”

We have specified the type of INFORS incubator

Line 207. I am not sure “transcriptionally” is the right work. I would suggest something like “.. by fusing recX in frame with the exo gene ..”

We use the term “transcriptional fusion” here to emphasize the fact that recX is not under control of its own, but a different (inducible) promoter in pKD46recX. In order to avoid confusions with translational fusions (that would result in a fusion protein) we would prefer to keep the term here.

There are a lot of gene names that are not italicized in the manuscript.  Also, E. coli is not italicized after the Results section.

This is a problem that obviously occurred in individual sections when transferring the article to the microorganisms template. We apologize for that and we have corrected this.

  1. Please change “..genes..” to read “.. of inserting the cat gene into the chromosome.”

We have done that.

Figure 1 legend. Remove “..in..” from “ ..in between..”

We have corrected this.

Line 256. I think the citation is not in superscript for “..mutations13.”  There are other places in the manuscript where this same error is occurring.

We thank the reviewer for noting this. We again apologize; this was not the case in the original manuscript and must have happened during transforming the manuscript into the microorganisms template. We have corrected this.

Faithfulness

We use now the word “accuracy”.

Line 282. Change the sentence to read “..we have also used pKD46recX to replace the stx2 gene of E .coli O104:H4 with cat.

We have corrected this.

Line 282. This result seems to be an afterthought. There is only one sentence for the result with no verification that stx2 gene was inactivated. This result is barely mentioned in the Discussion (line 344). I like that lab adapted strains were used to develop the method, but the real result here is the application of the method to a wild type EHEC strain. This should be one of the main points of the manuscript.

This is formally correct and one of the reasons why we also have used our method on a wild type EHEC strain (we have meanwhile also modified wild type EHEC O157:H7 and others). We have now added Supplementary Figure 6, which shows the results of the PacBio sequencing of the resulting strain. We did not detect stx2, nor did we detect cat aligning with other sequences than the regions up- and downstream stx2, suggesting a precise replacement of stx2 by cat in the resulting strain.

However, we do not believe that the stx gene itself is resulting in the overall reduced recombination efficiency in EHEC, but the prophage (induction) itself. Therefore, we consider it more important that the test strain contains an inducible phage than a stx gene (and is safe to work with). Nevertheless, it is of course necessary to show that the system also works in wild type EHEC strains.

Line 293.  “ultimately” should only be used once in the sentence.

We have changed that.

Line 307. Please indicate what fluorescent signal started to increase.

We have introduced “YFP” for clarification.

Line 317. The sentence reads awkwardly “..co-expression of recX apparently slightly, but significantly attenuated..” Either the result is significant or not.

It is clear that there is a small difference in between pKD46 and pKD46recX when PrecA is induced with sparfloxacin in the presence of arabinose. However, if this small difference is having a major impact (or “biological significance”) on the overall functionality of the sparfloxacin induced SOS response (given that the “real” response is much stronger when glucose is present in the medium, supplementary Figure 7), appears questionable to us and would require experimental verification.

Supplemental Figure 6. Please add E. coli K-12 MG1655 to the Figure or the Figure legend.

We have done this (new Supplementary Figure 7).

Lines 340-342. I missed how you determined there were no off-target mutagenic recombination. Did you sequence the recombinant strains?

We have not determined off-target mutagenic recombination events for our tests strain. However, we would have expected more false-positives in pKD46recX containing strains than in pKD46 containing strains, if pKD46recX would cause more off-target effects than pKD46. We did not observe that (Supplementary Figure 5).

However, E. coli O104:H4 stx2::cat was for example also fully sequenced (new supplementary figure 7).

Line 345. Change the sentence to read “..to delete the stx2 gene of the highly..”

We have done that.

Line 348.  Change “in” to “of”.

We have done that.

Line 365.  I think you can remove “prophage containing” from the sentence.

We have done that.

 Line 392. Remove “also” from the sentence.

We have done that.

Supplementary Figures 3 and 4.  Please add to the Figure legend what type of plot this is with the line representing the standard deviation and the box the average of the three replicates. Also, describe what the asterisks and “ns” indicate.

We have done that. We have also added the asterisks description in Figure 2 and Supplementary Figure 7.

Supplementary Figure 5 legend. Remove “in” before “between”

We have done that.